# Interventions for incarcerated adults with opioid use disorder in the United States: A systematic review with a focus on social determinants of health

Olivia K. Sugarman[1,2,3]*, Marcus A. Bachhuber[1,2], Ashley Wennerstrom[1,2,3], Todd Bruno[4], Benjamin F. Springgate[1,2,3]

1 Center for Healthcare Value and Equity, Louisiana State University Health Sciences Center–New Orleans, New Orleans, Louisiana, United States of America, 2 Section of Community and Population Medicine, Department of Medicine, School of Medicine, Louisiana State University Health Sciences Center–New Orleans, New Orleans, Louisiana, United States of America, 3 Department of Behavioral and Community Health Sciences, School of Public Health, Louisiana State University Health Sciences Center–New Orleans, New Orleans, Louisiana, United States of America, 4 Schwartz Law Firm, LLC, Mount Pleasant, South Carolina, United States of America

* okacsi@lsuhsc.edu

**Data Availability Statement:** All relevant data are within the manuscript.

## Abstract

Incarceration poses significant health risks for people involved in the criminal justice system. As the world's leader in incarceration, the United States incarcerated population is at higher risk for infectious diseases, mental illness, and substance use disorder. Previous studies indicate that the mortality rate for people coming out of prison is almost 13 times higher than that of the general population; opioids contribute to nearly 1 in 8 post-release fatalities overall, and almost half of all overdose deaths. Given the hazardous intersection of incarceration, opioid use disorder, and social determinants of health, we systematically reviewed recent evidence on interventions for opioid use disorder (OUD) implemented as part of United States criminal justice system involvement, with an emphasis on social determinants of health (SDOH). We searched academic literature to identify eligible studies of an intervention for OUD that was implemented in the context of criminal justice system involvement (e.g., incarceration or parole/probation) for adults ages 19 and older. From 6,604 citations, 13 publications were included in final synthesis. Most interventions were implemented in prisons (n = 6 interventions), used medication interventions (n = 10), and did not include SDOH as part of the study design (n = 8). Interventions that initiated medication treatment early and throughout incarceration had significant, positive effects on opioid use outcomes. Evidence supports medication treatment administered throughout the period of criminal justice involvement as an effective method of improving post-release outcomes in individuals with criminal justice involvement. While few studies included SDOH components, many investigators recognized SDOH needs as competing priorities among justice-involved individuals. This review suggests an evidence gap; evidence-based interventions that address OUD and SDOH in the context of criminal justice involvement are urgently needed.

**Funding:** The authors received no specific funding for this work. Schwartz Law Firm, LLC provided support in the form of salaries for authors [TB], but did not have any additional role in the study design, data collection and analysis, decision to publish, or preparation of the manuscript. The specific roles of these authors are articulated in the 'author contributions' section.

**Competing interests:** Schwartz Law Firm, LLC provided support in the form of salaries for authors [TB]. This commercial affiliation does not alter our adherence to PLOS ONE policies on sharing data and materials.

## Introduction

In the United States, the prison incarceration rate is the highest in the world at 655 per 100,000 [1]. Incarceration poses significant health risks for people involved in the criminal justice system [2–5]. Compared with the general population, incarcerated populations have much higher burdens of infectious diseases (e.g., hepatitis C virus, HIV, and tuberculosis) as well as mental illness and substance use disorder [6–10]. The transition from incarceration to the community itself is especially perilous [2,11,12]. In Washington State, for example, when compared with the general population, people reentering society from prison have a mortality rate nearly 13 times higher within the first two weeks post-release [3]. While multifactorial, this high mortality rate was driven largely by opioids, which were involved in approximately 1 in 8 post-release fatalities overall and over half of all overdose deaths [2,3]. Similar results were found in a more recent North Carolina study, in which the relative risk of opioid overdose death was 40 times higher than that of the general population within the first two weeks of release [12].

Increased risk of overdose post-release may be explained, at least in part, by decreased drug tolerance from a reduction in use or abstinence during incarceration. Returning to drug use following release may then be fatal due to the decreased tolerance level [2]. Medications for opioid use disorder (MOUD) for opioid use disorder, in the form of buprenorphine, methadone maintenance treatment, or extended-release injectable naltrexone (XR-NTX) reduce opioid misuse and overdose by reducing withdrawal symptoms and cravings through safe, controlled levels of medication [13]. Because of its efficacy, government agencies and national professional organizations recommend initiating MOUD upon incarceration and establishing continued treatment upon release [14–22].

Beyond MOUD treatment itself, social determinants of health (SDOH) are critical elements related to health outcomes post-release [21,23,24]. SDOH, as defined by the World Health Organization, are non-clinical factors including the "conditions in which people are born, grow, live, work and age. These circumstances are shaped by the distribution of money, power and resources at global, national, and local levels." [25]. Examples include housing, transportation, socioeconomic status. Addressing SDOH and attaining health care are often interrelated difficulties and conflicting priorities for formerly incarcerated people [21–23, 25–29]. Difficulty procuring employment, transportation or housing, for example, may pose immediate threats to well-being, making seeking health care services a lower priority [21,23,24,29–31]. The status or identifier of "formerly incarcerated" or "justice-involved" also severely restricts access to money, power, and resources. Many employment and housing applications require disclosing justice involvement, which may serve as a deterrent for potential employers, landlords, or loan officers, among others [29,30].

Previous systematic reviews have identified and compared studies of MOUD in prison settings and found treatment while incarcerated to be effective in potentially minimizing overdose risk [32]. Other studies have examined the impact of incarceration and social determinants of health on health outcomes, though we were unable to identify any systematic reviews [21,23,24,29–31]. Given the relationships between incarceration, OUD, and social determinants of health, evidence is urgently needed on intersectional interventions to improve outcomes for people who have a history of justice involvement and OUD.

To fill this gap, we conducted a systematic review of existing peer-reviewed literature describing interventions for justice-involved people with OUD through a social-determinants lens. The purpose of this systematic review is to 1) identify interventions for OUD that have been implemented as part of criminal justice system involvement, 2) determine which interventions also include a social determinants component, and 3) note any common elements between interventions with significant outcomes.

## Methods

We conducted a search of academic literature on May 6, 2019 to identify interventions for people with OUD implemented during incarceration following PRISMA standards for systematic reviews [33]. We used a broad definition of "incarceration" to include any involvement with the justice system. This includes prison, where people serve sentences greater than one year; jail, where people who have been arrested await trial or serve sentences less than one year; civil commitment, where people receive court-mandated inpatient treatment for a substance use disorder; probation and parole, where people serve their sentence in the community with regular check-ins to ensure adherence to sentence restrictions; and post-release, defined here as up to six months after being released from a jail or prison facility. A formal protocol for this review can be found at dx.doi.org/10.17504/protocols.io.69zhh76. Publication screening and selection was conducted by one team member (OS). Analysis was conducted by OS and TB.

We used PubMed to identify peer-reviewed articles. We limited publications to the last five years as drug overdose mortality peaked in 2014 [34], followed by declaration of opioid use as a public health emergency by the US Department of Health and Human Services in 2017 [35]. Grey literature and contact with study authors for additional studies were not pursued as part of this review. Further, because political context and region-specific legislation is particularly important for incarceration-related programming, non-U.S. based programs were not included in this review. We conducted all searches using a Boolean keyword search ((substance use OR medically assisted treatment OR opioid OR drug) AND (incarceration OR prison OR reentry OR jail)) in PubMed using the "best match" function. We completed a preliminary screen by removing duplicates and excluding articles that were not published in the last five years, were not published in English, did not have the full article text available, or did not include adults 19-years-old and older. We also searched ProQuest and Google Scholar using the same search terms and criteria. Publications identified using those methods were duplicates of the PubMed search and thus removed. Publications were limited to the last five years as drug overdose mortality peaked in 2014 [34], followed by declaration of opioid use as a public health emergency by the US Department of Health and Human Services in 2017 [35].

Next, we conducted a title and abstract screen to determine if publications fell within the inclusion criteria: 1) studies conducted in the U.S., 2) intervention studies only, 3) intervention studies for OUD, 4) for adults ages 19 and older. We excluded publications if: they described interventional studies that were conducted outside of the United States; the population of interest was under the age of 19; if studies were not interventional (e.g. epidemiological or surveillance studies); or did not investigate primary outcomes of interest. Primary outcomes of interest include: treatment initiation during incarceration, post-release opioid-related mortality, non-fatal overdose, and opioid use (heroin or prescription opioids), treatment initiation in community, adherence to treatment post-release, maintaining treatment post-release (i.e. keeping and attending appointments for treatment), and withdrawal symptoms. Finally, we reviewed the full text of the publications preliminarily meeting inclusion criteria to verify inclusion and relevance to this systematic review.

For the publications included in final review, the data were extracted individually by investigators and then compared. Findings were compiled in a categorical matrix (Table 1). Extracted data include: study and intervention characteristics, including target population, state, sample size, time of intervention implementation (intake, post-release, civil commitment, during incarceration, post-release, pre-release), implementation setting (jail, civil commitment facility, prison, transitions clinic), study design (case report, chart review, cohort, pilot study, randomized control trial), type of opioid intervention (buprenorphine, methadone, withdrawal management, XR-NTX, patient navigation, cross-sector collaboration),

**Table 1. Categorical matrix of systematic review findings.**

| Authors | State | Sample size | Time of intervention | Setting | Study design | Type of opioid intervention | Comparator | SDH included | Outcomes |
|---|---|---|---|---|---|---|---|---|---|
| Brinkley-Rubinstein et al. (2018) | RI | 223 | During incarceration | Prison | RCT[a] | MMT[b] | Forced Methadone withdrawal | For first appointment only<br>– Transportation<br>– Scheduling first MMT appointment<br>– Financial assistance | 12-month follow-up, MMT<br>– Heroin use less likely, prior 30 days ($p = 0.0467$)*<br>– Injection drug use less likely, prior 30 days ($p = 0.0033$)**<br>– Non-fatal overdose less likely (7% vs 18%, $p = 0.039$)*<br>– Continuous engagement with MMT during 12 month follow-up period* ($p = 0.0211$)* |
| Christopher et al. (2018) | MA | 318 | During civil commitment | Inpatient Civil Commitment | Prospective cohort | Civil commitment | - | None | Longer time to relapse positively associated with<br>– Keeping appointment for medication treatment following commitment ($p = 0.017$)* |
| Fox et al. (2014) | NY | 135 | Post-release | Transitions Clinic | Retrospective cohort | BT[c] | - | Offered for all clinic patients<br>– Social work referral<br>– Nutrition services<br>– Medicaid enrollment<br>– Health education<br>– Care coordination by formerly incarcerated community health worker | 6-month outcomes<br>– Fast median time from release to initial medical visit (10 days).<br>– Low care retention for opioid dependence (33%).<br>– Fewer buprenorphine-treated patients reduced opioid use (19%).<br>– Specifically cites need for SDH intervention and SDH as conflicting health priority. |
| Fresquez-Chavez & Fogger (2015) | NM | 55 | During incarceration | Jail | Case report | Withdrawal management (clonidine) | - | None | Withdrawal symptom scores (Subjective Opiate Withdrawal Scale)<br>– Baseline to 1 hour post-treatment ($p = .001$)***<br>– Baseline to 4 hours post-treatment ($p = .001$)*** |
| Gordon et al. (2014) | MD | 211 | Pre-release and Post incarceration | Prison | RCT, 2x2 factorial | In-prison treatment condition 1: BT while incarcerated Post-release service setting 1: Opioid treatment program post-incarceration | In-prison treatment condition 2: Counseling only while incarcerated Post-release service setting 2: Treatment at community health center post-incarceration | – Addressing barriers to community treatment entry (not specified)<br>– Employment<br>– Housing Offered in weekly group sessions provided by the study's addiction counselor | In-prison treatment condition<br>– Entering prison treatment more likely (99.0% v 80.4%, $p = .006$)**<br>– Community treatment entry (47.5% v 33.7%, $p = .012$)*<br>– Women more likely than men to complete prison treatment (85.7% v 52.7%, $p<0.001$).***<br>– 89.6% of all participants entered prison treatment<br>– 40.6% of all participants entered community all treatment<br>– 62.6% of all participants completed prison treatment |

*(Continued)*

**Table 1.** (Continued)

| Authors | State | Sample size | Time of intervention | Setting | Study design | Type of opioid intervention | Comparator | SDH included | Outcomes |
|---|---|---|---|---|---|---|---|---|---|
| Gordon et al. (2015) | MD | 27 | Pre-release | Prison | Pilot | XR-NTX[d] | - | – None | 9-month follow-up<br>– 77.8% of all participants completed prison injections<br>– 66.7% of all participants received first community injection<br>– 37% of all participants completed injection cycle<br>– Completers less likely to use opioids any time during the study vs non completers (p = 0.003).** |
| Gordon et al. (2017) | MD | 211 | Pre-release and Post incarceration | Prison | RCT, 2x2 factorial | In-prison treatment condition 1: Buprenorphine treatment while incarcerated Post-release service setting 1: Opioid treatment program post-incarceration | In-prison treatment condition 2: Counseling only while incarcerated Post-release service setting 2: Treatment at community health center post-incarceration | – Barriers to community treatment entry (not specified)<br>– Employment<br>– Housing<br>– Offered in weekly group sessions provided by the study's addiction counselor | 12 month follow-up Follow-up to Gordon (2014) In-prison treatment condition<br>– Higher mean number of days of community buprenorphine treatment v post-release medication initiation (p = .005)**<br>– No significant difference in negative urine opioid results of participants who entered community treatment. (p >0.14)<br>– No statistically significant effects for in-prison treatment condition for days of heroin use. (p >0.14) |
| Kobayashi et al. (2017) | RI | 107 | During incarceration | Prison | Pilot | Voluntary training, lay-person intranasal naloxone administration, opioid overdose prevention | - | – None | 1-month post-release follow-up<br>– 1 fatal opioid overdose (of 103 participants)<br>– 7 participants experienced non-fatal opioids<br>– 3 of 7 opioids ODs reversed using study-provided naloxone |
| Lee et al. (2015) | NY | 34 | Post-release | Jail | Randomized effectiveness trial | XR-NTX + counseling and referral intervention | Counseling and referral only | – None | 4-week post-release outcomes<br>– 15 of 17 participants initiated treatment<br>– Rates of opioid relapse 4 weeks post-release lower among XR-NTX participants (p<0.004, OR = .08, CI = 1.4–8.5)**<br>– More negative opioid urine samples in XR-NTX group (p<0.009, OR = 3.5, CI = 1.4–8.5)**<br>– No significant difference in rates of overdose<br>– No significant difference in participanion in other community drug treatment (19 v 12%)<br>[a]Small sample size<br>[a]Several measures relied on self-report |
| Morse et al. (2017) | NY | 200 | Post-release | Transitions Clinic | Chart review | BT | - | SDOH included in the Transitions Clinic model, but not measured for this chart review. | – Thirty (70%) of the 38 women in sample with opioid use disorder received methadone or suboxone. |

*(Continued)*

**Table 1.** (Continued)

| Authors | State | Sample size | Time of intervention | Setting | Study design | Type of opioid intervention | Comparator | SDH included | Outcomes |
|---|---|---|---|---|---|---|---|---|---|
| Prendergast, McCollister, & Warda (2017) | CA | 732 | During Incarceration | Jail | RCT | SBIRT[e] | Drug and alcohol, HIV risk information + program list of local providers | – None | – No significant difference in change in opioid risk between SBIRT and control group (p = 0.13)<br>– No significant difference in attending outpatient treatment, past 12 months (p = 0.49)<br>– No significant difference for any primary or secondary outcomes between groups. |
| Rich et al. (2015) | RI | 223 | Intake | Prison | RCT | Continued MMT post-release | Methadone taper | Transportation, Scheduling<br>– Financial assistance<br>– With first methadone treatment appointment only | 1 month post-release follow-up<br>– Of participants assigned to continued MMT post-release, 97% (n = 111) attended community methadone clinic vs. 71% (n = 77) of participants assigned to methadone taper (p<0.0001)***<br>– MMT participants twice as likely to return to community methadone clinic within 1 month post-release (Hazard risk = 2.04, 95% CI = 1.48–2.80)<br>– N = 1 mortality (Continued MMT group), no significant difference<br>– N = 1 non-fatal overdose in continued MMT group, n = 2 in methadone taper group (p = 0.423) |
| Vocci et al. (2015) | MD | 104 | During Incarceration | Prison | RCT | BT | No BT | – None | 10 weeks post-therapy initiation<br>62% of participants (n = 63) remained on BT at release from prison<br>– 50% of participants completed 10 weeks of treatment (n = 60).<br>– Suggest that buprenorphine administered to non-opioid-tolerant adults should be started at a lower, individualized dose than customarily used for adults actively using opioids. |

[a] RCT = randomized controlled trial

[b] MMT = methadone Maintenance Treatment

[c] BT = buprenorphine treatment

[d] XR-NTX = injectable extended-release naltrexone

[e] Screening, brief intervention, and referral to treatment

[*] p ≤ 0.05

[**] p ≤ 0.01

[***] p ≤ 0.001

comparator, whether and how SDOH were addressed in the intervention (e.g. support for housing, transportation, financing medical care, nutrition services, and case management or social services referral to navigate SDOH issues), and study outcomes. Not all outcomes were available for each study.

## Results

In the initial keyword search in PubMed, 6,604 citations were identified. After applying filters, 993 publications met the preliminary screen. From those, we identified 45 full-text articles through the abstract and title screen. Finally, through full review, we identified 13 publications that met all inclusion criteria (Fig 1).

Of the 32 publications removed from consideration, 14 were removed because they described studies that were not interventions, six were not implemented as part of criminal justice involvement, seven were not opioid-specific, one was not exclusively for people who are involved in the criminal justice system, and three were removed because the outcomes

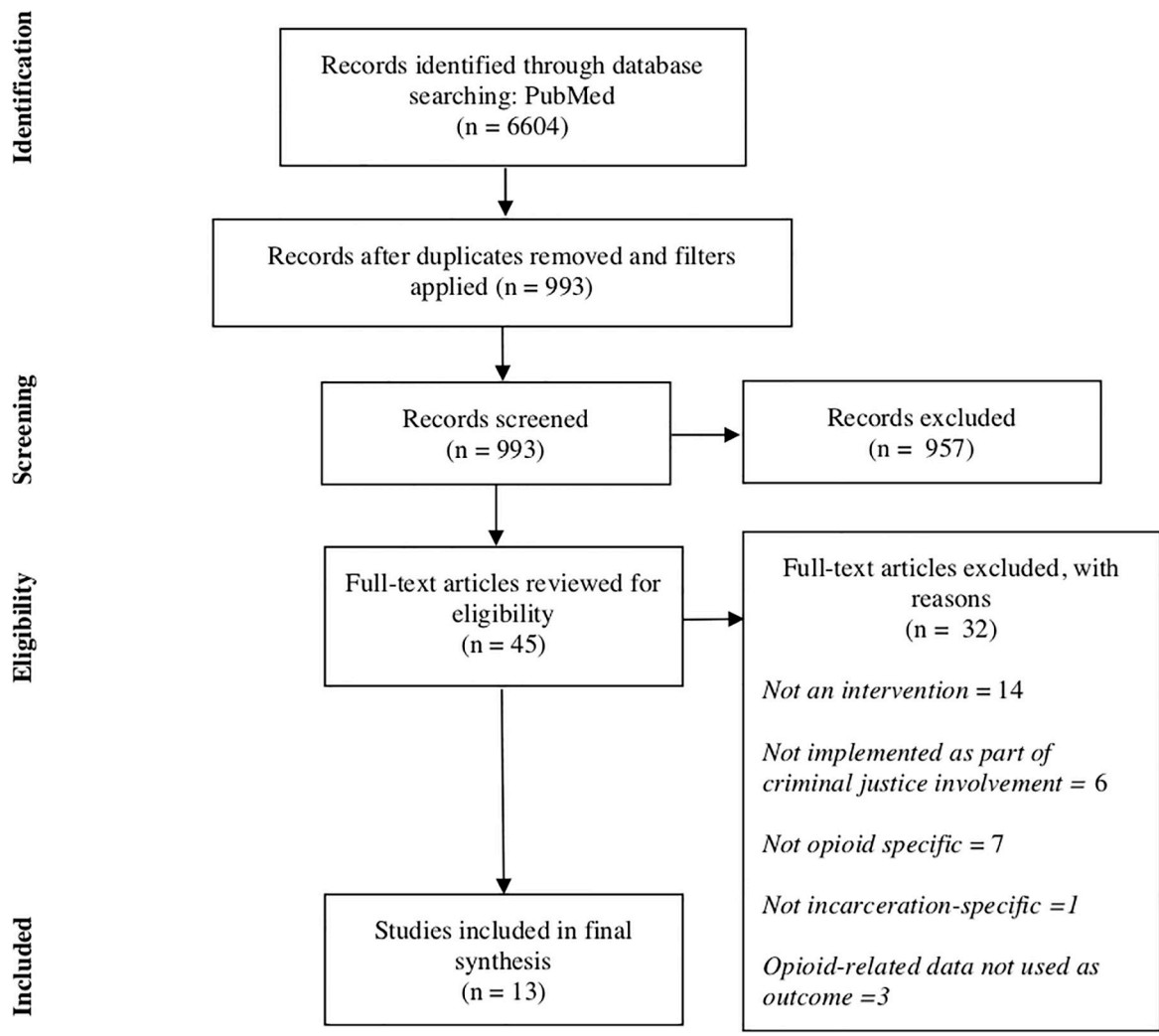

**Fig 1. PRISMA Systematic Review Diagram.** Adapted from:Moher D, Liberati A, Tetzlaff J, Altman DG, The PRISMA Group (2009). *P*referred *R*eporting *I*tems for *S*ystematic Reviews and *M*eta-*A*nalyses: The PRISMA Statement. PLoS Med 6(7): e1000097. doi:10.1371/journal.pmed1000097.

measured did not meet inclusion criteria. Fig 1 provides additional details in a PRISMA diagram. Of the 13 publications included for final synthesis, some included continuation studies, leaving 12 distinct interventions.

The majority of interventions were implemented in prisons (n = 6 interventions, 7 publications) [36–42] and jails (n = 3) [43–45]. The remainder were implemented in Transitions Clinics (n = 2) [46,47] or in a civil commitment facility (n = 1) [48]. Results are described in Table 1 and tabulated in Table 2.

Interventions primarily involved evidence-based medication treatments (n = 9 interventions, 10 publications) [36–39,41–44,46,47] the majority of which utilized buprenorphine (n = 4 interventions, 5 publications) [37,39,42,46,47], methadone (n = 2)[36,41], or (XR-NTX) (n = 2) [38,44]. One intervention used withdrawal management with clonidine as a non-opioid method of aiding newly incarcerated people who use opioids in a New Mexico county jail [43]. There was a distinction between XR-NTX studies and other pharmacological interventions. XR-NTX improved outcomes, though XR-NTX is administered only immediately prior to release rather than during incarceration [38,44].

Two studies focused on opioid overdose fatality prevention including a pilot of a voluntary intranasal naloxone administration [38] and training for people incarcerated in a Rhode Island prison [40]. The only non-pharmaceutical intervention study examined the effects of Screening, Brief Intervention, and Referral to Treatment (SBIRT) for OUD [45].

Three of the twelve interventions included social determinants-related components as part of either the study design or implementation [36,37,39,41,46]. Several publications alluded to SDOH as a barrier to receiving care, but only three provided any social determinants-related support as part of the intervention. One intervention offered transportation, scheduling assistance, and financial assistance for participants' first methadone treatment appointment post-incarceration [36,41]. Another intervention offered counseling on barriers to community treatment entry, employment post-incarceration, and housing post-incarceration in weekly group sessions provided by the study's addiction counselor [37,39]. The third study described SDOH support programs offered to all patients of the Transitions Clinic intervention, which included: referrals to social work services, nutrition services, Medicaid enrollment, health education, and care coordination by a formerly incarcerated community health worker [46].

Interventions that included evidence-based medication treatments (i.e., buprenorphine, methadone, XR-NTX) yielded improvements in outcomes of interest, especially in studies that measured post-incarceration connection to community treatment and continuation of treatment [36–39,41–44,46,47]. Significance of results for health outcomes was fairly consistent across medication types (methadone, buprenorphine, XR-NTX), though time of treatment initiation was associated with intervention success. In general, the effectiveness and long-term impact of methadone and buprenorphine treatment interventions on non-fatal overdose, overdose mortality, post-release opioid use, and seeking and maintaining treatment post-incarceration were associated with early initiation during incarceration and consistent treatment during incarceration [36–39,42–44].

Relative to controls, one intervention (SBIRT) yielded no significant difference in outcomes. Another, a Transitions Clinic found that care retention and opioid use reduction were low and specifically cited a need for social determinants support as part of care, as many of their patients had competing social determinants-related priorities [46].

## Discussion

In a systematic review of the evidence, we identified a range of evidence-based options to support people with OUD who are incarcerated or recently released from incarceration in the U.

**Table 2. Tabulated results of systematic review categorical matrix, by number of publications and interventions.**

| Variable | Publications n | Interventions n |
|---|---|---|
| **State** | | |
| California | 1 | 1 |
| Maryland | 4 | 3 |
| Massachusetts | 1 | 1 |
| New Mexico | 1 | 1 |
| New York | 3 | 3 |
| Rhode Island | 3 | 3 |
| **Time of intervention** | | |
| Civil commitment | 1 | 1 |
| Intake | 1 | 1 |
| During Incarceration | 5 | 5 |
| Pre-release | 1 | 1 |
| Post-release | 3 | 3 |
| Pre- and Post-release | 2 | 1 |
| **Implementation setting** | | |
| Inpatient civil commitment facility | 1 | 1 |
| Jail | 3 | 3 |
| Prison | 7 | 6 |
| Transitions Clinic | 2 | 2 |
| **Study design** | | |
| Case report | 1 | 1 |
| Chart review | 1 | 1 |
| Retrospective cohort | 1 | 1 |
| Prospective cohort | 1 | 1 |
| Pilot study | 2 | 2 |
| Randomized control trial | 6 | 5 |
| Randomized effectiveness trial | 1 | 1 |
| **Type of opioid intervention** | | |
| Buprenorphine Treatment | 5 | 4 |
| Civil commitment | 1 | 1 |
| Clonidine withdrawal management | 1 | 1 |
| Extended-release Naltrexone (XR-NTX) | 2 | 2 |
| Methadone maintenance treatment | 2 | 2 |
| Screening, Brief Intervention, and Referral to Treatment | 1 | 1 |
| XR-NTX training | 1 | 1 |
| **Social Determinants of Health** | | |
| Addressed* | 5 | 5 |
| Not addressed | 8 | 8 |
| Housing, employment, barriers to treatment | 2 | 1 |
| Social work referral, nutrition services, Medicaid enrollment, health education, care coordination | 1 | 1 |
| Barriers to community treatment entry, employment, housing | 2 | 1 |

Number of publications and interventions differ as two publications described outcomes of the same intervention at different follow-up periods.

S. In reviewed studies, MOUD had significant beneficial impacts on outcomes when treatment was initiated early in criminal justice system involvement and maintained throughout incarceration. While several interventions did integrate social determinants components, these

were included in only a minority of interventions reviewed. Results of studies presented in this review is consistent with the current evidence-base regarding MOUD and incarceration, and SDOH as a potential barrier to good health outcomes post-release. However, this review reveals that a gap at the intersection of MOUD, incarceration, and SDOH persists. There is a substantial opportunity to incorporate SDOH into interventions to support the health and well-being of critically at-risk populations who are incarcerated or have been recently released.

Mass incarceration and the opioid epidemic are simultaneously salient crises, but are often considered separately from one another. As criminal justice reform and the opioid epidemic converge in national policy discourse, U.S. policy-makers must support and fund rigorous research and programmatic evaluation to identify methods of addressing SDOH to support OUD treatment among justice-involved people. Altogether, implementing policy and evidence-based programs that simultaneously prioritize SDOH management and OUD treatment is paramount to narrowing the health and social disparities supported by mass incarceration of the last 40 years in the U.S.

Studies included in this review reported clinical interventions typically using medication-based treatments. However, new studies are implementing non-clinical strategies to fortify both interpersonal and cross-sectoral relationships. Such non-clinical strategies may serve as a complementary solution to medication treatment either in carceral facilities with policies that restrict MOUD options such as buprenorphine or post-release. For instance, the Bronx Transitions Clinic has proposed several new initiatives to complement current services [46]. Such programs include a peer-mentorship program and support groups to encourage positive coping skills [46].

For cross-sectoral relationships, the MAT Implementation in Community Correctional Environments (MATICCE) study sought to strengthen referral and treatment continuation relationships through corrections-community partnerships [49]. MATICCE tested implementation strategies for connecting correctional agencies and incarcerated people approaching release with evidence-based treatment services that already existed in their communities [49]. MATICCE established 20 Department of Corrections (DoC)-community dyads in 11 states, which were then tasked with creating ways of making and fortifying inter-organizational relationships and familiarizing Department of Corrections staff with MOUD [49]. This approach simultaneously avoided expanding agencies' responsibilities, facilitated alignment of state and facility policies, and encouraged dyads to create their own solutions to building inter-organizational relationships. Though results were mixed, future studies with inter-agency collaboration designs may refine on this first iteration. Further work may establish additional evidence-informed collaborative alternatives to complement more prevalent corrections-only rehabilitative programming. Bolstering community capacities and establishing and fortifying existing community-based services may enhance both the community and the long-term success of formerly incarcerated people.

## Limitations

This review has several limitations. We may not have identified some pilot programs initiated by county, state, or federal departments of corrections, health departments, or community organizations because we searched only the academic literature. This review does not include programs currently implemented by respective criminal justice systems or facilities. Some existing interventions may not have publicly available evaluations. Further, carceral facilities and systems can vary significantly, even within the same county or state and so studies may not be generalizable to other settings.

## Recommendations

Based on this systematic review, we recommend that future interventions for OUD among justice-involved people specifically include attention to understanding and addressing the impacts of SDOH on post-incarceration health outcomes. We further recommend implementing process and outcomes evaluations for new incarceration-based or post-incarceration programs to address OUD. We strongly suggest that formerly incarcerated individuals, particularly those who have been treated successfully for OUD, participate in program design and evaluation to maximize potential utility and end-user relevance.

Recent changes in state legislature and federal discourse have started to address the intersections of OUD and social determinants among justice-involved people [15–17, 19–21]. Future studies should assess the impacts of innovative state-level programming for OUD treatment among formerly incarcerated people. Additionally, to better understand current and best practices, future efforts should focus on describing the national landscape of available OUD and social determinants programs as well as their compatibility for mutual integration.

## Conclusion

This systematic review of interventions for OUD implemented as part of US criminal justice system involvement synthesized results from several innovative pilot programs and study interventions. The interest in opioid-specific programs and interventions for people involved in the criminal justice system is rising, but more research is needed to understand the key role that addressing SDOH could play in contributing to improved health outcomes. The existing evidence base suggests that medication treatments such as buprenorphine and methadone should administered early in incarceration and continued for the duration of incarceration, particularly for those in prison. Although SDOH were frequently noted as a potential competing priority to engaging in treatment, few interventions to-date have addressed SDOH in the intervention or study design. Those that did include SDOH cited competing priorities as a major determinant of treatment initiation and adherence. Through individual-level interventions or building strong cross-sector collaborations, future interventions for incarcerated people with OUD should integrate medication treatments with interventions to address social determinants of health.

## Supporting information

**S1 Table. PRISMA checklist.** Completed PRISMA Checklist.
(DOC)

## Author Contributions

**Conceptualization:** Olivia K. Sugarman, Marcus A. Bachhuber, Ashley Wennerstrom, Benjamin F. Springgate.

**Data curation:** Olivia K. Sugarman, Todd Bruno.

**Supervision:** Benjamin F. Springgate.

**Writing – original draft:** Olivia K. Sugarman.

**Writing – review & editing:** Olivia K. Sugarman, Marcus A. Bachhuber, Ashley Wennerstrom, Todd Bruno, Benjamin F. Springgate.

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
