## [Decision Letter · Decision Letter 0]

5 Sep 2019

PONE-D-19-17598

Interventions for incarcerated adults with opioid use disorder in the United States: A systematic review with a focus on social determinants of health

PLOS ONE

Dear Ms. Sugarman,

Thank you for submitting your manuscript to PLOS ONE. After careful consideration, we feel that it has merit but does not fully meet PLOS ONE’s publication criteria as it currently stands. Therefore, we invite you to submit a revised version of the manuscript that addresses the points raised during the review process.

Reviewer concerns highlight some discrepancies between the studies as presented and summarized and the evidence, in particular with respect to naltrexone. In addition there is a significant lack of detail on the methods, which make interpretation of the findings of the study difficult in context of the literature reviewed.  Finally the emphasis on social determinants is not fully justified -- the predominant interventions for this issue are related to MAT. In fact there is now legislation in several states mandating the continuation of MAT for those incarcerated. 

We would appreciate receiving your revised manuscript by Oct 20 2019 11:59PM. To enhance the reproducibility of your results, we recommend that if applicable you deposit your laboratory protocols in protocols.io, where a protocol can be assigned its own identifier (DOI) such that it can be cited independently in the future. For instructions see: http://journals.plos.org/plosone/s/submission-guidelines#loc-laboratory-protocols

We look forward to receiving your revised manuscript.

Kind regards,

Becky L. Genberg

Academic Editor

PLOS ONE

Journal Requirements:

2. Thank you for stating the following in the Competing Interests section: "The authors have declared that no competing interests exist."

We note that one or more of the authors are employed by a commercial company:Todd Bruno Law company.

Reviewers' comments:

Reviewer's Responses to Questions

**Comments to the Author**

1. Is the manuscript technically sound, and do the data support the conclusions?

Reviewer #1: Yes

Reviewer #2: Yes

2. Has the statistical analysis been performed appropriately and rigorously? 

Reviewer #1: Yes

Reviewer #2: N/A

3. Have the authors made all data underlying the findings in their manuscript fully available?

Reviewer #1: Yes

Reviewer #2: Yes

4. Is the manuscript presented in an intelligible fashion and written in standard English?

Reviewer #1: Yes

Reviewer #2: Yes

5. Review Comments to the Author

Reviewer #1: Generally good work.

Line comments

18: insert “post-release” before “fatalities”

31: the studies reviewed include those delivered to persons on probation. Individuals on probation might never have been incarcerated. Perhaps say “throughout the period of criminal justice involvement” instead of “through incarceration” which parallels the language in lines 21-22 as to the actual scope of your study.

32: again, insert “post-release” before “outcomes”

34: delete “participants” and substitute “justice involved individuals” or an equivalent phrase since the studies cover both formerly incarcerated individuals and probationers and parolees.

47: insert “or justice-involved” after “formerly incarcerated”

54: insert “after release” after the word “overall”

59: change “from” to “through”

99: delete “the raw” and change to “these”

Table 1: review entries under Summary of Findings and fix any language errors. E.g., Rich 2015 line 2 should likely read “participants assigned to MMT attended” or whatever acronym is used in the article for methadone maintenance therapy.

148-156: The paragraph appears to accurately reflect what the individual studies showed. However, XR-NTX does not have as wonderful a track record as implied in the Gordon and particularly the Lee studies. Lee’s November 2017 study – outside the scope of the review – tells a truer story than his 2015 study (which, should be noted, was funded in part by Alkermes, the aggressive drug maker that is spending a lot of money to make sure it’s drug is the drug of choice for prison and other justice settings. They have successfully lobbied to state laws changed so that drug courts can only offer XR-NTX to participants. As administrators at places like RI DOC know, where all three approved MOUD treatments are offered, most individuals prefer Buprenorphrine. Substantially fewer choose XR-NTX. Which is the conclusion of Josh’s 2017 study, available at https://www.thelancet.com/journals/lancet/article/PIIS0140-6736(17)32812-X/fulltext. Take a look at the summary of findings there. Mike Gordon’s study is more robust, but has a small N (37). In any event, the last sentence of this paragraph doesn’t apply to XR-NTX since it is always and only administered immediately prior to release.

225-227: It is crystal clear that addressing SDH is critically important to successful post-release reintegration. Housing, employment, family and community reconnection, etc.: all present competing reentry and survival needs that often trump health needs, including recovery and treatment for SUD. All of the articles about Transitions Clinics address this fact. In any event, I would change the second part of this sentence to read something like: “but more research is needed to understand the key role that addressing SDH could play in contributing to long-term recovery and improved health outcomes….”

227-229: This sentence may be true, but not for XR-NTX.

Reviewer #2: The manuscript (MS) addresses the important topic of opioid use disorder (OUD) among incarcerated adults. The MS is, for the most part well-written but, requires additional explication of rationale and methods (see below).

First, the Introduction highlights the problem of mass incarceration in the U.S., the high rates and commonly fatal outcomes of untreated OUD among those incarcerated, and the impact of incarceration on social determinants of health (SDH). However, the MS does not refer to any – or whether there have been any – reviews already conducted on these topics. Identifying other relevant reviews (if any), their findings, and how the current study may add to this literature would aid in identifying a rationale for this study.

This reviewer finds the MS’s treatment of intervention “outcomes” most problematic. Outcomes are vaguely defined throughout the MS. The purpose of the review (stated on p. 5, ln. 63) does not specify outcomes of interest. The Methods section only states that “a summary of findings” (p. 6, ln. 95) were extracted from eligible studies. There is no indication of how study outcomes were considered in determining study eligibility. This contributes to considerable confusion when reading on page seven (ln 111) that one study was removed from the review “because opioid-related measures were not used as an outcome” and again in the Results section (p. 13, ln. 149) that both opioid use-related outcomes and justice-related outcomes were evaluated. Continuing with this concern, on the same page (ln. 164), the Discussion summarizes that this review found “in reviewed studies, medication treatments for OUD had significant beneficial impacts on outcomes when…” Outcomes should again be specified here.

While mentioned under Limitations (p. 15), the Methods section should explicitly indicate that the grey literature or contact with study authors for additional studies were not pursued as part of this review.

The Methods section does not provide any information with which readers can determine the reliability of data extraction. Were data extracted independently by investigators and then compared? Was a data extraction tool/form used?

The Methods section indicates only studies published within the last five years were eligible for study inclusion. It is unclear why this five-year period was chosen (why not four years or seven years or other?).

Table 1 should provide follow-up periods evaluated among the included studies.

It is unclear why the Discussion section chooses to highlight the MATICCE study when, according to Table 1, opioid use-related outcomes and justice-related outcomes were not reported as findings from that study (see comment related to Outcomes above).

The Discussion would benefit from a summary of study findings on the strength of current evidence on the topic reviewed.

6. PLOS authors have the option to publish the peer review history of their article (what does this mean?). If published, this will include your full peer review and any attached files.

Reviewer #1: No

Reviewer #2: No

---

## [Author Response · Author response to Decision Letter 0]

20 Oct 2019

Responses to the Editors:

Reviewer concerns highlight some discrepancies between the studies as presented and summarized and the evidence, in particular with respect to naltrexone. In addition there is a significant lack of detail on the methods, which make interpretation of the findings of the study difficult in context of the literature reviewed. 

-We thank the reviewers for these comments and agree that clarification will be helpful to readers. We made several revisions to respond to these particular issues, described in detail below. 

Finally the emphasis on social determinants is not fully justified -- the predominant interventions for this issue are related to MAT. In fact there is now legislation in several states mandating the continuation of MAT for those incarcerated. 

-We agree that additional clarification and justification of the emphasis on social determinants is warranted and we have added it to the introduction.

-Revised - Beyond MOUD treatment itself, social determinants of health (SDOH) are critical elements related to health outcomes post-release [23–25]. SDOH, as defined by the World Health Organization, are non-clinical factors including the “conditions in which people are born, grow, live, work and age. These circumstances are shaped by the distribution of money, power and resources at global, national, and local levels.” [26] Examples include housing, transportation, socioeconomic status. Addressing SDOH and attaining health care are often interrelated difficulties and conflicting priorities for formerly incarcerated people [23–25, 27–29]. Difficulty procuring employment, transportation or housing, for example, may pose immediate threats to well-being, making seeking health care services a lower priority [23–25, 30–32]. The status or identifier of “formerly incarcerated” or “justice-involved” also severely restricts access to money, power, and resources. Many employment and housing applications require disclosing justice involvement, which may serve as a deterrent for potential employers, landlords, or loan officers, among others [30, 31]. (p.4 lines 75-87).

To enhance the reproducibility of your results, we recommend that if applicable you deposit your laboratory protocols in protocols.io, where a protocol can be assigned its own identifier (DOI) such that it can be cited independently in the future. For instructions see: http://journals.plos.org/plosone/s/submission-guidelines#loc-laboratory-protocols

-Thank you for connecting us to this resource. We have deposited our protocol in protocols.io. The protocol can be found at: dx.doi.org/10.17504/protocols.io.69zhh76.

-Style and formatting changes have been made per the PLOS ONE requirements. 

2. Thank you for stating the following in the Competing Interests section: "The authors have declared that no competing interests exist."

We note that one or more of the authors are employed by a commercial company: Todd Bruno Law company.

-Todd Bruno is the sole proprietor of Todd Bruno Law, LLC. At the time of this set of revisions, Todd Bruno is now employed by and affiliated with Schwartz Law Firm, LLC. Neither of these entities have any commercial interests in this manuscript’s topic. The following statement has been added to the Funding Statement and included in the cover letter accompanying this submission.

The authors received no specific funding for this work.

Schwartz Law Firm, LLC provided support in the form of salaries for authors [TB], but did not have any additional role in the study design, data collection and analysis, decision to publish, or preparation of the manuscript. The specific roles of these authors are articulated in the ‘author contributions’ section.

Within your Competing Interests Statement, please confirm that this commercial affiliation does not alter your adherence to all PLOS ONE policies on sharing data and materials by including the following statement: "This does not alter our adherence to PLOS ONE policies on sharing data and materials.” (as detailed online in our guide for authors http://journals.plos.org/plosone/s/competing-interests) .

-The Competing Interests Statement, updated in the cover letter, contains the statement “This does not alter our adherence to PLOS ONE policies on sharing data and materials.”

-A Supporting Information heading and accompanying captions are now included at the end of the manuscript. (p.20 line 295)

Reviewer #1: Generally good work.

-We thank the reviewer for the kind words.

Line comments

18: insert “post-release” before “fatalities”

-Revised – “opioids contribute to nearly 1 in 8 post-release fatalities overall” (Line 36)

31: the studies reviewed include those delivered to persons on probation. Individuals on probation might never have been incarcerated. Perhaps say “throughout the period of criminal justice involvement” instead of “through incarceration” which parallels the language in lines 21-22 as to the actual scope of your study.

-Revised – “Evidence supports medication treatment administered throughout the period of criminal justice involvement…” (Line 47-48)

32: again, insert “post-release” before “outcomes”

-Revised – “…as an effective method of improving post-release outcomes in individuals with criminal justice involvement.” (Lines 48-49)

34: delete “participants” and substitute “justice involved individuals” or an equivalent phrase since the studies cover both formerly incarcerated individuals and probationers and parolees.

-Revised – “While few studies included SDOH components, many investigators recognized SDOH needs as competing priorities among justice-involved individuals.” (Lines 49-51).

47: insert “or justice-involved” after “formerly incarcerated”

-Revised – “The status or identifier of “formerly incarcerated” or “justice-involved” also severely restricts access to money, power, and resources.” (Lines 84-85)

54: insert “after release” after the word “overall”

-Revised to include “post-release” as above – “While multifactorial, this high mortality rate was driven largely by opioids, which were involved in approximately 1 in 8 post-release fatalities overall and over half of all overdose deaths [2,3]. (Lines 61-63).

59: change “from” to “through”

-Revised – “To fill this gap, we conducted a systematic review of existing peer-reviewed literature describing interventions for justice-involved people with OUD through a social-determinants lens. (Lines 95-97)

99: delete “the raw” and change to “these”

-Revised – “These After applying filters, 993 publications met the preliminary screen. From those, we identified 45 full-text articles through the abstract and title screen.” (Lines 155-156).

Table 1: review entries under Summary of Findings and fix any language errors. E.g., Rich 2015 line 2 should likely read “participants assigned to MMT attended” or whatever acronym is used in the article for methadone maintenance therapy.

-Please see the revised Table 1.

148-156: The paragraph appears to accurately reflect what the individual studies showed. However, XR-NTX does not have as wonderful a track record as implied in the Gordon and particularly the Lee studies. Lee’s November 2017 study – outside the scope of the review – tells a truer story than his 2015 study (which, should be noted, was funded in part by Alkermes, the aggressive drug maker that is spending a lot of money to make sure it’s drug is the drug of choice for prison and other justice settings. They have successfully lobbied to state laws changed so that drug courts can only offer XR-NTX to participants. As administrators at places like RI DOC know, where all three approved MOUD treatments are offered, most individuals prefer Buprenorphrine. Substantially fewer choose XR-NTX. Which is the conclusion of Josh’s 2017 study, available at https://www.thelancet.com/journals/lancet/article/PIIS0140-6736(17)32812-X/fulltext. Take a look at the summary of findings there. Mike Gordon’s study is more robust, but has a small N (37). In any event, the last sentence of this paragraph doesn’t apply to XR-NTX since it is always and only administered immediately prior to release.

-Thank you for this important information. We have now included a short statement clarifying the differences between XR-NTX and other medication management options. 

-“There was a distinction between XR-NTX studies and other pharmacological interventions. XR-NTX improved outcomes, though XR-NTX is administered only immediately prior to release rather than during incarceration [39,45].” (Lines 183-185)

225-227: It is crystal clear that addressing SDH is critically important to successful post-release reintegration. Housing, employment, family and community reconnection, etc.: all present competing reentry and survival needs that often trump health needs, including recovery and treatment for SUD. All of the articles about Transitions Clinics address this fact. In any event, I would change the second part of this sentence to read something like: “but more research is needed to understand the key role that addressing SDH could play in contributing to long-term recovery and improved health outcomes….”

-Revised – “The interest in opioid-specific programs and interventions for people involved in the criminal justice system is rising, but more research is needed to understand the key role that addressing SDOH could play in contributing to improved health outcomes..” (Lines 284-286)

227-229: This sentence may be true, but not for XR-NTX.

-Revised – “The existing evidence base suggests that medication treatments such as buprenorphine and methadone should administered early in incarceration and continued for the duration of incarceration, particularly for those in prison.”

Responses to Reviewer 2:

The manuscript (MS) addresses the important topic of opioid use disorder (OUD) among incarcerated adults. The MS is, for the most part well-written but, requires additional explication of rationale and methods (see below).

First, the Introduction highlights the problem of mass incarceration in the U.S., the high rates and commonly fatal outcomes of untreated OUD among those incarcerated, and the impact of incarceration on social determinants of health (SDH). However, the MS does not refer to any – or whether there have been any – reviews already conducted on these topics. Identifying other relevant reviews (if any), their findings, and how the current study may add to this literature would aid in identifying a rationale for this study. 

-We agree with the reviewer that this additional background information would be helpful for readers. We have now revised the introduction to include additional detail. We identified few reviews on these topics. Their findings are included in the text. 

-Previous systematic reviews have identified and compared studies of MOUD in prison settings and found treatment while incarcerated to be effective in potentially minimizing overdose risk [33]. Other studies have examined the impact of incarceration and social determinants of health on health outcomes, though we were unable to identify any systematic reviews [23–25, 30–32]. Given the relationships between incarceration, OUD, and social determinants of health, evidence is urgently needed on intersectional interventions to improve outcomes for people who have a history of justice involvement and OUD. (Lines 88-94)

This reviewer finds the MS’s treatment of intervention “outcomes” most problematic. Outcomes are vaguely defined throughout the MS. The purpose of the review (stated on p. 5, ln. 63) does not specify outcomes of interest. The Methods section only states that “a summary of findings” (p. 6, ln. 95) were extracted from eligible studies. There is no indication of how study outcomes were considered in determining study eligibility. This contributes to considerable confusion when reading on page seven (ln 111) that one study was removed from the review “because opioid-related measures were not used as an outcome” and again in the Results section (p. 13, ln. 149) that both opioid use-related outcomes and justice-related outcomes were evaluated. Continuing with this concern, on the same page (ln. 164), the Discussion summarizes that this review found “in reviewed studies, medication treatments for OUD had significant beneficial impacts on outcomes when…” Outcomes should again be specified here.

-Thank you for bringing this to our attention. Outcomes have been clarified and defined in the Methods section. We also attempted to clarify inclusion and exclusion criteria.

We excluded publications if: they described interventional studies that were conducted outside of the United States; the population of interest was under the age of 19; if studies were not interventional (e.g. epidemiological or surveillance studies); or did not investigate primary outcomes of interest. Primary outcomes of interest include: treatment initiation during incarceration, post-release opioid-related mortality, non-fatal overdose, and opioid use (heroin or prescription opioids), treatment initiation in community, adherence to treatment post-release, maintaining treatment post-release (i.e. keeping and attending appointments for treatment), and withdrawal symptoms. (Lines 132-139)

While mentioned under Limitations (p. 15), the Methods section should explicitly indicate that the grey literature or contact with study authors for additional studies were not pursued as part of this review.

-A statement reflecting the absence of grey literature is now included in the Methods section:

“Grey literature and contact with study authors for additional studies were not pursued as part of this review.” (Lines 117-118)

The Methods section does not provide any information with which readers can determine the reliability of data extraction. Were data extracted independently by investigators and then compared? Was a data extraction tool/form used? 

-“For the publications included in final review, the data were extracted individually by investigators and then compared. Findings were compiled in a categorical matrix (Table 1).” (Lines 142-143)

Further, a protocol for this review was developed and published on interventions.io to provide additional clarity in identifying texts, data extraction, and analysis.

The Methods section indicates only studies published within the last five years were eligible for study inclusion. It is unclear why this five-year period was chosen (why not four years or seven years or other?). 

-Thank you for bringing our attention to this. A statement clarifying the selection was added in the Methods section.

Publications were limited to the last five years as drug overdose mortality peaked in 2014 [35], followed by declaration of opioid use as a public health emergency by the US Department of Health and Human Services in 2017 [36].” (Lines 127-129)

Table 1 should provide follow-up periods evaluated among the included studies.

-Added to Table 1, please see revised table. 

It is unclear why the Discussion section chooses to highlight the MATICCE study when, according to Table 1, opioid use-related outcomes and justice-related outcomes were not reported as findings from that study (see comment related to Outcomes above).

-The MATICCE study did not meet inclusion criteria; thank you for pointing this out to us. It has been removed from analysis and the resulting table, figures, and analysis. However, we did want to highlight the MATICCE study as a non-clinical systems level approach to connecting people to treatment post-incarceration. 

-“Studies included in this review reported clinical interventions typically using medication-based treatments. However, new studies are implementing non-clinical strategies to fortify both interpersonal and cross-sectoral relationships. Such non-clinical strategies may serve as a complementary solution to medication treatment either in carceral facilities with policies that restrict MOUD options such as buprenorphine or post-release. For instance, the Bronx Transitions Clinic has proposed several new initiatives to complement current services [47]. Such programs include a peer-mentorship program and support groups to encourage positive coping skills [47]. 

For cross-sectoral relationships, the MAT Implementation in Community Correctional Environments (MATICCE) study sought to strengthen referral and treatment continuation relationships through corrections-community partnerships [50]…” (Lines 235-245)

The Discussion would benefit from a summary of study findings on the strength of current evidence on the topic reviewed.

-Thank you for this suggestion. Please see the revision and addition to the first paragraph of the discussion.

“In a systematic review of the evidence, we identified a range of evidence-based options to support people with OUD who are incarcerated or recently released from incarceration in the U.S. In reviewed studies, MOUD had significant beneficial impacts on outcomes when treatment was initiated early in criminal justice system involvement and maintained throughout incarceration. While several interventions did integrate social determinants components, these were included in only a minority of interventions reviewed. Results of studies presented in this review is consistent with the current evidence-base regarding MOUD and incarceration, and SDOH as a potential barrier to good health outcomes post-release. However, this review reveals that a gap at the intersection of MOUD, incarceration, and SDOH persists. There is a substantial opportunity to incorporate SDOH into interventions to support the health and well-being of critically at-risk populations who are incarcerated or have been recently released.” (Lines 216-226)

---

## [Decision Letter · Decision Letter 1]

6 Jan 2020

Interventions for incarcerated adults with opioid use disorder in the United States: A systematic review with a focus on social determinants of health

PONE-D-19-17598R1

Dear Dr. Sugarman,

We are pleased to inform you that your manuscript has been judged scientifically suitable for publication and will be formally accepted for publication once it complies with all outstanding technical requirements.

With kind regards,

Becky L. Genberg

Academic Editor

PLOS ONE

Additional Editor Comments (optional):

Reviewers' comments:

Reviewer's Responses to Questions

**Comments to the Author**

1. If the authors have adequately addressed your comments raised in a previous round of review and you feel that this manuscript is now acceptable for publication, you may indicate that here to bypass the “Comments to the Author” section, enter your conflict of interest statement in the “Confidential to Editor” section, and submit your "Accept" recommendation.

Reviewer #2: All comments have been addressed

Reviewer #3: All comments have been addressed

2. Is the manuscript technically sound, and do the data support the conclusions?

Reviewer #2: (No Response)

Reviewer #3: Yes

3. Has the statistical analysis been performed appropriately and rigorously? 

Reviewer #2: (No Response)

Reviewer #3: N/A

4. Have the authors made all data underlying the findings in their manuscript fully available?

Reviewer #2: (No Response)

Reviewer #3: Yes

5. Is the manuscript presented in an intelligible fashion and written in standard English?

Reviewer #2: (No Response)

Reviewer #3: Yes

6. Review Comments to the Author

Reviewer #2: (No Response)

Reviewer #3: The authors have been responsive to the reviewers' critiques. Recommend accept. One small change to consider: the RI department of corrections is a statewide system that does not have a distinction between jail/prison. So, in the table when characterizing the study settings as jail/prison--Ri doesn't really fall into either completely.

7. PLOS authors have the option to publish the peer review history of their article (what does this mean?). If published, this will include your full peer review and any attached files.

Reviewer #2: No

Reviewer #3: Yes: Lauren Brinkley-Rubinstein

---

## [Editor Report · Acceptance letter]

10 Jan 2020

PONE-D-19-17598R1 

Interventions for incarcerated adults with opioid use disorder in the United States: A systematic review with a focus on social determinants of health 

Dear Dr. Sugarman:

I am pleased to inform you that your manuscript has been deemed suitable for publication in PLOS ONE. Congratulations! Your manuscript is now with our production department. 

With kind regards,

on behalf of

Dr. Becky L. Genberg 

Academic Editor

PLOS ONE